# Eph and Ephrin function in dispersal and epithelial insertion of pigmented immunocytes in sea urchin embryos

Oliver A Krupke[1], Ivona Zysk[2], Dan O Mellott[2], Robert D Burke[2]*

[1]Department of Biology, University of Victoria, Victoria, Canada; [2]Department of Biochemistry and Microbiology, University of Victoria, Victoria, Canada

**Abstract** The mechanisms that underlie directional cell migration are incompletely understood. Eph receptors usually guide migrations of cells by exclusion from regions expressing Ephrin. In sea urchin embryos, pigmented immunocytes are specified in vegetal epithelium, transition to mesenchyme, migrate, and re-enter ectoderm, distributing in dorsal ectoderm and ciliary band, but not ventral ectoderm. Immunocytes express Sp-Eph and Sp-Efn is expressed throughout dorsal and ciliary band ectoderm. Interfering with expression or function of Sp-Eph results in rounded immunocytes entering ectoderm but not adopting a dendritic form. Expressing Sp-Efn throughout embryos permits immunocyte insertion in ventral ectoderm. In mosaic embryos, immunocytes insert preferentially in ectoderm expressing Sp-Efn. We conclude that Sp-Eph signaling is necessary and sufficient for epithelial insertion. As well, we propose that immunocytes disperse when Sp-Eph enhances adhesion, causing haptotactic movement to regions of higher ligand abundance. This is a distinctive example of Eph/Ephrin signaling acting positively to pattern migrating cells.

## Introduction

**\*For correspondence:** rburke@ uvic.ca

**Competing interests:** The authors declare that no competing interests exist.

Dispersal of cells from a site of origin to form a predictable pattern throughout an organism is a spectacular demonstration of directed migration, an essential component of the metazoan patterning toolkit. The complex movements of neural crest or the extensions of central nervous system axons are familiar vertebrate examples that are brought about by a relatively small number of cellular mechanisms. Attraction to a chemical source, permissive substrates, and mechanisms of exclusion or repulsion appear to the principal means of controlling even complex cell migrations (*Davies, 2005*).

In many situations Ephrin and Eph are important components of the signaling that regulates migration of cells (*Pasquale, 2005*, *2008*; *Xu and Henkemeyer, 2012*; *Poliakov et al., 2004*). Eph receptors are a family of receptor tyrosine kinases that are activated by cell surface ligands, Ephrins. Ephrins are bound to membranes through a glycosylphosphatidylinositol linkage (Ephrin A) or a transmembrane domain (Ephrin B). The receptors and ligands are found throughout metazoans and they are thought to accompany the evolution of multicellularity (*Srivastava et al., 2010*; *Tischer et al., 2013*). Vertebrate receptors and ligands are diverse with as many as 16 Eph receptors and 8 Ephrins. The principal developmental functions of Eph/Ephrin signaling include defining tissue domains and guiding migrating cells or growth cones (*Klein, 2012*; *Cayuso et al., 2015*). In the best-understood models, Eph receptors repulse cells from regions expressing Ephrin, or with reverse signaling, where ligand bearing cells respond to receptor binding, Ephrin-expressing cells are repulsed from cells expressing Eph receptors. Eph and Ephrins regulate cytoskeletal dynamics and often integrate activity of other receptor ligand systems to coordinate adhesive and migratory responses to the environment of the cell (*Bashaw and Klein, 2010*; *Bush and Soriano, 2012*).

**eLife digest** During animal development, numerous cells move around the embryo to form and shape the growing tissues. As these cells move, they are guided to their destination by molecular cues. The embryo's tissues produce these cues and the cues can either repel or attract migrating cells. Ephrins are a large and well-studied family of proteins that serve as guidance cues and are found on the surface of certain types of cells. Some migrating cells have receptors for Ephrin and are repelled from tissues that contain Ephrin proteins. In these cases, the repulsive interaction between Ephrins and cells with receptors ensures that migrating cells avoid certain locations and reach the correct final destination.

The sea urchin is an important model organism for studying how animals develop and in particular how genes control animal development. This is in part because these animals can be easily manipulated in the laboratory and are more closely related to animals with backbones than many other model organisms. Sea urchins also have a relatively simple set of genes; many of which are similar to the human form of the gene. In sea urchin embryos, pigmented cells called immunocytes are known to migrate from one region of the embryo to another where they form part of its immune system. However it was not clear what guides this migration.

Sea urchins produce one type of Ephrin protein and its associated receptor, and now Krupke et al. show that immunocytes carry the receptor for Ephrin and migrate to embryonic tissues that produce high levels of this Ephrin. This finding suggested that the Ephrin is actually attracting the immunocytes to their final destination rather than repelling them. Further experiments supported this idea and revealed that immunocytes that lack the Ephrin receptor fail to enter the right tissue. Similarly, altering the pattern of Ephrin in the embryo's tissues altered immunocyte migration in a predictable way.

These findings of Krupke et al. suggest that Ephrin and its receptor have changed their biological functions during evolution of animals. This raises a number of questions for future research including whether the molecular mechanisms used by Ephrin and its receptor to attract immunocytes in sea urchins is the same as that used to repel cells in other species.

In vertebrates Eph and Ephrins have complex patterns of expression and interaction, which complicates functional studies. Genomic studies of sea urchins reveal that there is a single Eph receptor (Sp-Eph) and a single Ephrin ligand (Sp-Efn), making them an attractive model for studies of function (*Whittaker et al., 2006*). In addition, urchins are basal deuterostomes, so they provide an opportunity to study the range of morphogenetic mechanisms employed by embryos that share a common ancestor with vertebrates (*Drescher, 2002*; *Mellott and Burke, 2008*). In *S. purpuratus*, embryos express Sp-Eph and Sp-Efn in two overlapping domains of ectoderm and Eph/Ephrin signaling appears to have a role in morphogenesis of the ciliary band (*Krupke and Burke, 2014*).

Pigmented immunocytes are a well-studied cell lineage in urchin embryos because of their distinctive reddish granules and their early specification within the nested rings of endomesodermal cells in the vegetal plate. Specification of pigmented immunocyte precursors begins in cleavage stages when micromeres express Delta, which initiates Notch-mediated specification of a non-skeletogenic mesoderm domain in an adjacent ring of veg2-derived blastomeres (*Sherwood and McClay, 1999*; *Sweet et al., 2002*). This domain is first demarcated by expression of the transcription factor glial cells missing (GCM), which is critical to the specification of pigmented immunocytes in the dorsal half of the domain (*Ransick and Davidson, 2006*, *2012*; *Ruffins and Ettensohn, 1996*). During gastrulation, the pigmented immunocytes transition to mesenchyme and migrate to the ectoderm (*Gibson and Burke, 1985*, *1987*; *Takata and Kominami, 2004*). Once in the ectoderm the cells adopt a distinctive dendritic form and mediate innate responses to pathogens (*Solek et al., 2013*). Although many aspects of the lineage have been well studied (*Barsi et al., 2015*; *Beeble and Calestani, 2012*; *Buckley and Rast, 2015*), we know little about the dispersal and insertion into epithelium of pigmented immunocytes.

Here we report immunocyte precursors express the receptor, Sp-Eph, while migrating over a gradient of Sp-Efn ligand on the basal surfaces of dorsal ectodermal cells. Manipulations of Sp-Eph and

Sp-Efn concentrations reveal that suppression of Sp-Eph signaling interferes with insertion of immunocytes into the ectoderm. Altering ligand concentration indicates that pigmented immunocytes are more likely to insert in ectoderm expressing high levels of Sp-Efn. We propose Sp-Eph and Sp-Efn function initially in the dispersion of immunocytes precursors, and subsequently they are necessary for the insertion of mesenchyme into epithelium.

## Results

### Immunocyte migration

Pigmented immunocytes of sea urchin embryos arise in the vegetal mesoderm and begin to transition from epithelium to mesenchyme during the initial phase of archenteron invagination (*Gibson and Burke, 1985*; *Ruffins and Ettensohn, 1996*) (*Video 1*). The precursors migrate to the ectoderm where they insert between epithelial cells (*Videos 2, 3*). In their migratory phase, immunocytes are spherical and have numerous short, spike-like projections. When immunocytes insert into epithelium, they extend lamellipodia that spread distally and project numerous filopodia (*Video 4*). Live imaging indicates that when pigmented immunocytes insert into epithelium, they become stationary or move only small distances. However, they continue to actively extend and retract filopodia (*Videos 2, 5*). As pigmented immunocytes in the ectoderm remain relatively stationary, we concluded that they select a site to insert during their migration from the vegetal plate and that insertion includes a change in immunocyte morphology.

### Immunocyte patterning correlates with the distribution of Ephrin

The distribution of the pigmented immunocytes is stereotypic in that they do not enter ventral ectoderm and the cells are most dense at the tips of larval arms (*Figure 1A*). As well, the density of immunocytes is graded in the dorsal ectoderm (*Figure 1F*). Pigmented immunocytes are most dense at the tip of larval body and along the edge of the ciliary band. Counts of pigmented immunocytes in a series of 5 midline quadrants arrayed along the axis from the animal pole to the vertex indicate a parabolic distribution (*Figure 1G*).

Sp-Efn is expressed in the dorsal ectoderm and ciliary band (*Krupke and Burke, 2014*). A more detailed examination of the ligand indicates an apparent correlation between the relative fluorescence, or immunoreactivity of ectoderm expressing Sp-Efn and the distribution of pigmented immunocytes. Sp-Efn is not expressed in ventral ectoderm, but is expressed in the ciliary band and dorsal ectoderm and it appears to be most abundant at the tips of the larval arms (*Figure 1B–E*). Measurements of fluorescence intensity per pixel in 5 evenly spaced quadrants along the axis from the animal pole domain to the vertex indicate that there is a gradient of Sp-Efn abundance in the dorsal ectoderm (*Figure 1H*). These observations show a correlation in the distribution of pigmented immunocytes and the abundance of Sp-Efn, suggesting a role for Sp-Efn in localization of these cells.

**Video 1.** Epithelial-mesenchyme transition (EMT) and migration of DGP:GFP labelled pigmented immunocytes. Live imaging of an embryo (40 hr) injected with DGP:GFP (*Ransick and Davidson, 2012*). GFP is expressed in pigmented immunocyte precursors undergoing EMT (first arrow). Subsequent to this cell migrating to the ectoderm, another cell (second arrow) undergoes EMT and also attaches to the ectoderm. The duration of the sequence is 90 min.

### Sp-Efn is expressed on cytonemes

Fixation with PEM, a buffer designed to stabilize cytoskeletal structure (*Vielkind and Swierenga, 1989*) results in immunolocalization of Sp-Efn on the basal surface of ectodermal cells (*Figure 2A*). In addition to small cytoplasmic granules, Sp-Efn

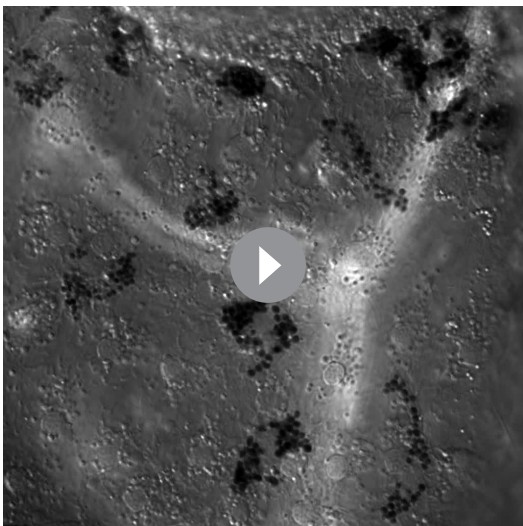

**Video 2.** Pigmented immunocytes remain at the site of insertion for prolonged periods of time. Live imaging of the pigmented immunocytes in an early pluteus. The cells remain active, but they are not displaced from their location throughout the 1.5 hr sequence. As pigmented immunocytes remain relatively stationary, we concluded that they select a site to insert during the phase of migration subsequent to release from the vegetal plate.

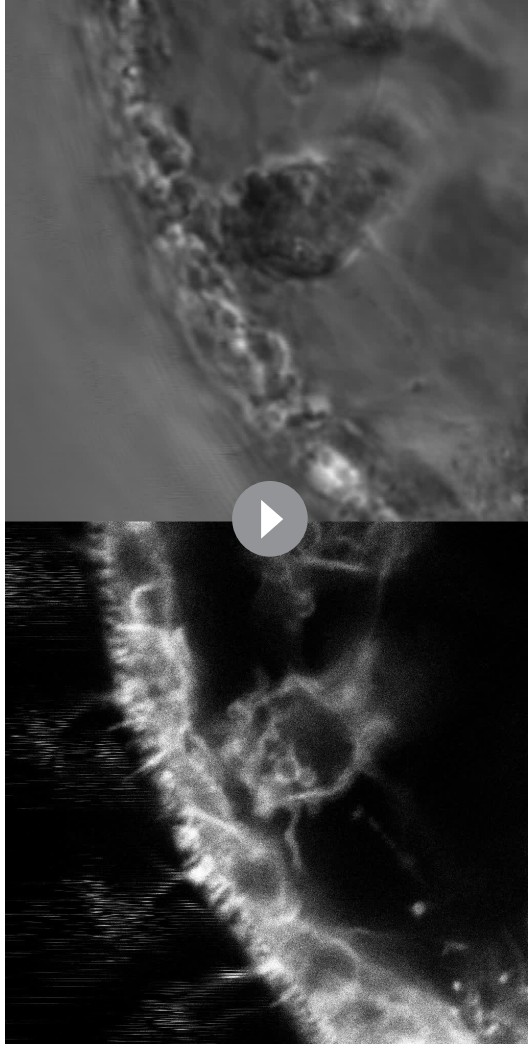

**Video 3.** Pigmented immunocyte inserting into ectoderm. In the insertion sequence a cell first approaches the ectoderm, then inserts between epithelial cells (30 min sequence).

is abundant on long, thin projections that radiate into the blastocoel (*Figure 2B*). The processes are unbranched, up to 15 µm in length and 0.1–0.2 µm in diameter (*Figure 2C*). Notably, fixation with ice-cold methanol, or buffered formaldehyde solutions, localizes Sp-Efn to dorsal ectodermal cells, but does not preserve the basal processes. Localization with apical markers, spectrin or actin, and DIC imaging to establish the position of the basal surface, confirms that Sp-Efn is abundant on processes radiating into the blastocoel (*Figure 2D–J*). Filopodial projections from ectoderm in urchin embryos have been described (*Andrews, 1897*; *Dan, 1960*; *Gustafson and Wolpert, 1961*; *Vacquier, 1968*, *Miller et al., 1995*). Co-localization of Sp-Efn with filamentous actin indicates that the processes contain actin (*Figure 2H–J*). In addition, we prepared embryos for live imaging by expressing a GFP construct containing 80 nucleotides from the C-terminus of mammalian RAS (membraneGFP), which is known to localize fluorescence to membranes (*Moriyoshi et al., 1996*). In sequential images captured at a rate of 4 frames per minute, fluorescent processes emanating from the basal surfaces of ectodermal cells are apparent (*Figure 2K,L*, *Video 6*). Similarly, when embryos express the actin-binding Lifeact-GFP (*Riedl et al., 2008*), thin fluorescent projections emanate from the bases of ectodermal cells (*Figure 2M,N*, *Video 7*). The projections are common on the basal surfaces of dorsal and ventral ectoderm, but not associated with endodermal epithelia. We concluded from these studies that Sp-Efn protein is presented on membrane-delimited, actin-containing, cytonemal processes on the basal surfaces of ectodermal cells.

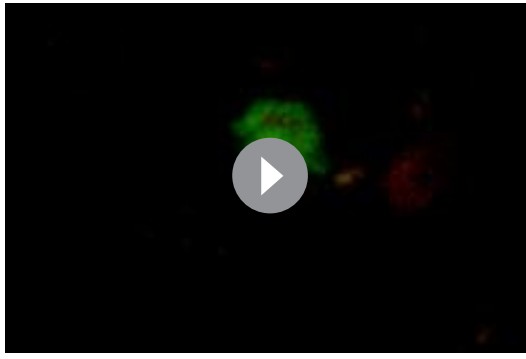

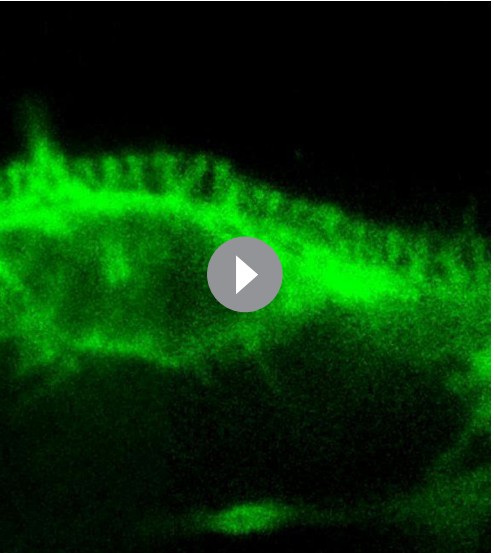

**Video 4.** Pigmented immunocyte inserts into ectoderm and changes from migratory form to a form similar to a dendritic cell. The sequence is 90 min long, beginning with a 48 hr embryo.

**Video 5.** In this sequence, a pigmented immunocyte that is inserted in the ectoderm extends a single process through the ectoderm to the outside of the embryo (arrow). The process extends and retracts repeatedly throughout this 30 min sequence. Note also the distinctive process extended from the blastocoelar surface of the immunocyte.

## Immunocytes express Sp-Eph

In hatched blastulae, cells in the vegetal plate that are immunoreactive with the immunocyte-specific antibody Sp1 (*Gibson and Burke, 1985*), are also immunoreactive with anti-Sp-Eph (*Krupke and Burke, 2014*) (*Figure 3A–C*). Sp-Efn is present at the same time on the basal surfaces of ectodermal cells (*Figure 3D*). By 36 hr, presumptive pigmented immunocytes are mesenchyme within the blastocoel and continue to express Sp-Eph (*Figure 3E–G*). In gastrulae (42 hr), pigmented immunocytes have migrated to the basal surface of dorsal and ciliary band ectoderm (*Figure 3H–J*) and they extend projections into the ectoderm. The projections can extend beyond the apical ectodermal surface (insets *Figure 3H–J*, *Video 5*). In early plutei (72 hr), pigmented immunocytes have a typical dendritic morphology in which cells have multiple lamellipodia-like projections between ectodermal cells immediately beneath apical junctions (*Figure 3K–M*). The pigmented immunocytes continue to express Sp-Eph in early pluteus stages. These findings indicate Sp-Eph is expressed by pigmented immunocyte from the time of their release from the vegetal plate throughout early larval development.

## Eph signaling is necessary for immunocyte migration

To determine the function of Eph/Ephrin signaling in pigmented immunocyte development, we perturbed Eph/Ephrin signaling with an Eph kinase inhibitor (NVP, *Martiny-Baron et al., 2010*) or anti-sense morpholinos (Sp-Eph MO1). Suppressing expression of Sp-Eph produces embryos that gastrulate and have defects in the ciliary band (*Krupke and Burke, 2014*) (*Figure 4A*). Immunocyte precursors migrate to the ectoderm and at 40 hr there is no difference in the number of Sp1 immunoreactive cells between control and experimental treatments (*Figure 4A,E*). However at 72 hr pigmented immunocyte abundance is significantly reduced in NVP treated (p<0.0001) or Sp-Eph MO1 injected (p = 0.009) embryos (*Figure 4E*). In 72 hr embryos injected with Sp-EphMO1, the immunocytes are larger in diameter, with fewer lamellipodia and shorter filopodia than embryos injected with a control morpholino (*Figure 4F*). As well, in embryos treated with NVP or derived from eggs injected with Sp-Eph MO1, a significantly larger proportion of immunocytes are within the blastocoel (*Figure 4G*). Thus, interfering with Eph signaling results in immunocytes that are impaired in their ability to insert into the ectoderm and do not transition to the characteristic dendritic morphology. In addition, immunocytes that remain in the blastocoel of Sp-Eph MO injected embryos commonly have fragmented nuclei and reduced Sp1 intensity. Using an antibody that detects a cleaved or activated form of caspase-3 (a marker for apoptosis), we find that there are a small number of pigmented immunocytes that are cleaved caspase-3 positive in 55 hr Sp-Eph MO injected embryos

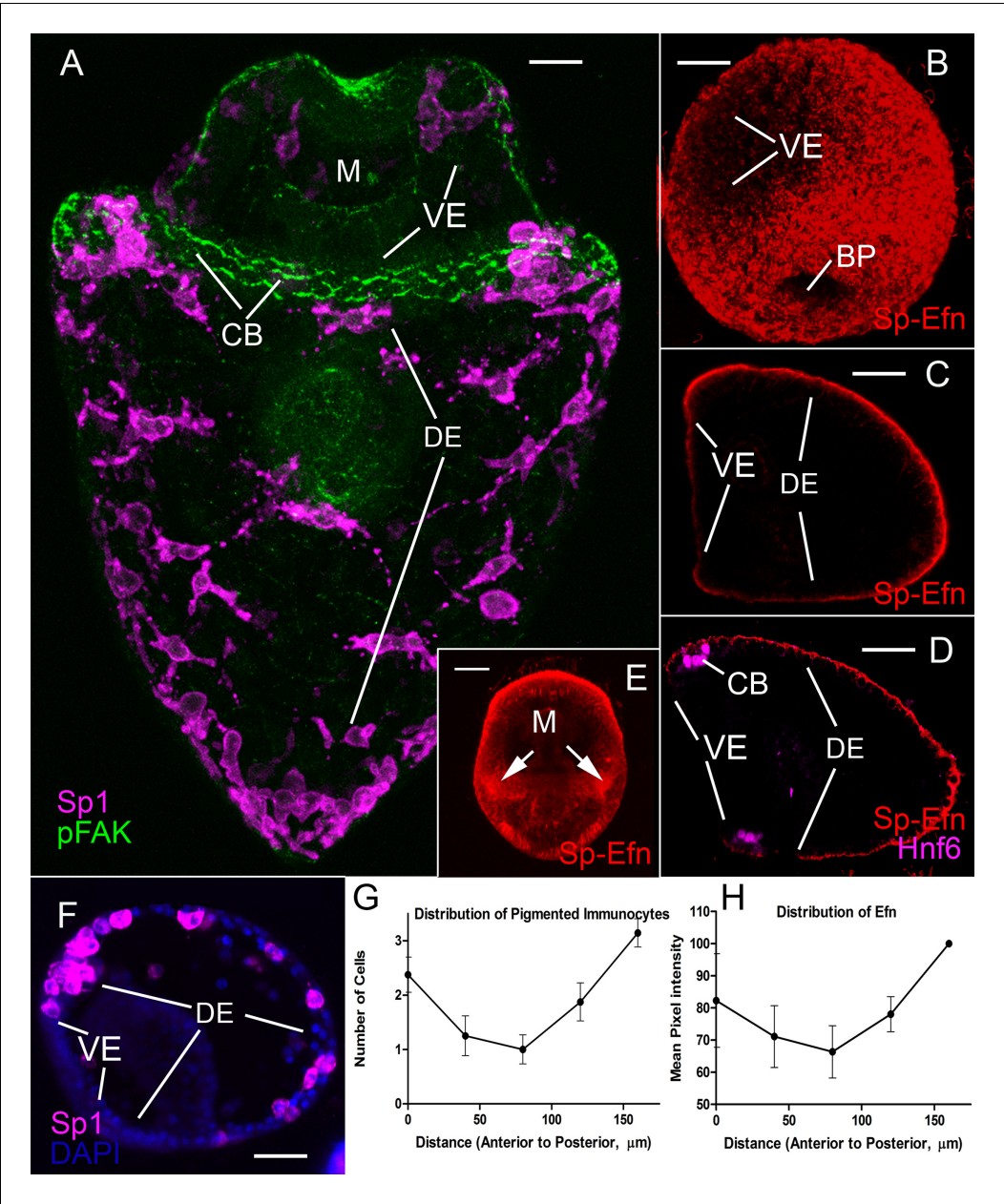

**Figure 1.** The distribution of pigmented immunocytes correlates with the abundance of Sp-Efn. (**A**) Maximum intensity projection of confocal optical sections of an early pluteus larva prepared with Sp1, a pigmented immunocyte cell surface marker, and pFAK, a marker for apical junctions that are prominent in ciliary band cells (MeOH fixation). Pigmented immunocytes are scattered throughout the dorsal ectoderm (DE), which is the larval surface posterior to the ciliary band (CB). Immunocytes do not normally insert in the ventral ectoderm (VE), which is the larval surface surrounding the mouth (M) and encircled by the ciliary band. Immunocytes are most dense at the tips of larval arms, along the edge of the ciliary band, and the posterior tip of the larval body. Bar = 15 µm (**B**) Maximum intensity projection of confocal optical sections of a gastrula prepared with an antibody to Sp-Efn. The ligand is not expressed in the ventral ectoderm (VE). BP, blastopore (PFA fixation) Bar = 20 µm (**C**) Single sagittal optical section of a prism larva showing the expression of Sp-Efn only in the dorsal ectoderm and ciliary band and not in the ventral ectoderm (VE) (PEM fixation) Bar = 20 µm (**D**) Single sagittal optical section of an early pluteus larva showing the expression of Sp-Efn and the ciliary band marker, Hnf6. Sp-Efn expression is not uniform throughout the dorsal ectoderm, there is an apparent gradation of expression that is highest at the vertex of the larva. (PFA fixation). Bar = 20 µm (**E**) Maximum intensity projection of an early pluteus prepared with anti-Sp-Efn and oriented with the ventral surface foremost. Arrows indicate high levels of immunoreactivity where the postoral larval arms are situated. M, mouth (PEM fixation) Bar = 20 µm (**F**) Mid sagittal optical section of an early pluteus

*Figure 1 continued on next page*

*Figure 1 continued*

prepared with Sp1 to show the distribution of pigmented immunocytes (MeOh fixation). Bar = 20 μm (**G**) The distribution of pigmented immunocytes in the ectoderm between the animal pole domain and the vertex was determined from a set of mid-sagittal images from 6 embryos prepared with Sp1 (see *Figure 1—figure supplement 1*). The distance from the vertex of the larva along the surface to the animal pole domain was divided into five zones each 40 μm long and the number of pigmented immunocytes in each zone was counted. Mean and S.E.M. (**H**) The distribution of Sp-Efn in the same region of ectoderm was determined from a set of mid-sagittal images from 6 embryos prepared with anti-Sp-Efn (See *Figure 1—figure supplement 1*). Projections of 6-image stacks were prepared and equal-sized rectangles were positioned at 40 μm intervals along the ectoderm. Mean intensity per pixel, normalized to the highest intensity per embryo was determined within each rectangle and Mean and S.E.M. plotted.

The following source data and figure supplement are available for figure 1:

**Source data 1.** Source data for *Figure 1G and H*.

**Figure supplement 1.** *Figure 1G* Quantification of pigmented immunocytes.

(*Figure 4B–D,H*). This supports a model in which pigmented immunocytes with suppressed Eph signaling are less able to insert in the ectoderm, remain rounded in form, and a small number become apoptotic.

## Ectopic expression of Sp-Efn alters the distribution of pigmented immunocytes

To determine the effects of ectopic Ephrin expression on pigment cell distribution we analyzed embryos from eggs injected with Sp-Efn RNA. In 72 hr control larvae (eggs injected with GFP RNA), GFP is expressed in all ectoderm and pigmented immunocytes are distributed as in uninjected embryos. In 72 hr embryos from eggs injected with Sp-Efn RNA, the ventral ectoderm is immunoreactive with anti-Sp-Efn (*Figure 4L,M*). In these embryos pigmented immunocytes insert in the ventral ectoderm (*Figure 4I–K*) but not endodermal or mesodermal epithelia. These data indicate that ectopic Sp-Efn is sufficient to mislocalize pigmented immunocytes to ventral ectoderm.

To further examine the effects of altering the levels of expression of Sp-Efn, we injected one blastomere of a 2-cell embryo to create mosaic embryos in which one half of the embryo is expressing the ectopic gene. The left-right axis is not fixed in *S. purpuratus*, so embryos produced in this manner have random positioning of the injected half (*Henry et al., 1992*). To track the position of the half injected we co-injected GFP and assessed the number of pigmented immunocytes associated with each half of the embryo. When half of the embryo expresses Sp-Efn ectopically there are more immunocytes inserted in ectoderm on the injected half of the embryo than on the uninjected half or the GFP injected controls (*Figure 5A–C,M*, *Video 8*). When half of the embryo has Sp-Efn expression suppressed with a morpholino (Sp-Efn MO2), more immunocytes insert on the un-injected half than in the half containing the morpholino, or the GFP control (*Figure 5D–F,M*). Many of the immunocytes in the half expressing GFP and containing Sp-Efn MO2 insert close to the interface of the two halves and extend processes to the Efn expressing side (*Figure 5E*). To test the combined effect of the morpholino and the RNA, we injected eggs with Sp-Efn MO2 and one blastomere with Sp-Efn RNA. When expression of Sp-Efn is suppressed throughout the embryo and half of the embryo overexpresses Sp-Efn, nearly all of the pigmented immunocytes insert in ectoderm of the half of the embryo expressing Sp-Efn (*Figure 5G–I,M*). Control embryos expressed only GFP in half of the embryo and pigmented immunocytes were evenly distributed (*Figure 5J–L,M*). We concluded from these experiments that pigmented immunocytes are more likely to insert in the regions of ectoderm that express Sp-Efn and that high levels of expression of Sp-Efn enhances pigment cell insertion.

## Discussion

Ephrins and Eph receptors are typically expressed on cell surfaces and signaling is mediated by direct cellular contact (*Klein, 2012*). Soluble Ephrins have been shown to function as competitive

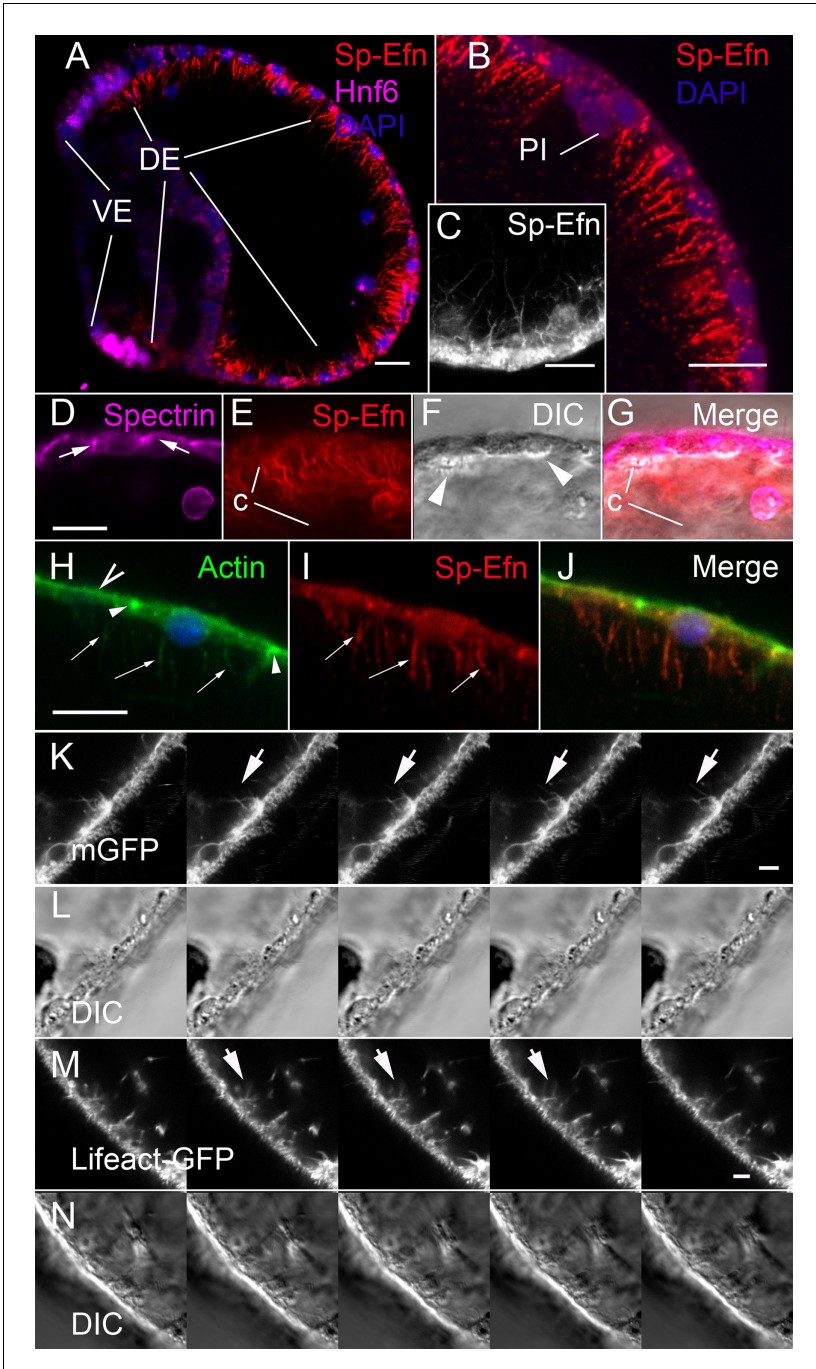

**Figure 2.** Sp-Efn is expressed on membrane-bounded, actin-containing cytonemes on the blastocoelar surface of dorsal ectoderm. (**A**) Fixation that preserves the cytoskeleton (PEM, *Vielkind and Swierenga, 1989*) reveals that Sp-Efn immunoreactivity localizes to long thin projections from the basal surfaces of dorsal ectoderm (**DE**). VE, ventral ectoderm Bar = 15 μm. (**B,C**) Higher magnification showing cellular detail of basal surface of dorsal ectodermal cells. PI, pigmented immunocyte. (**B**) Z-projection of 5 optical sections of a 72 hr larva. C Single optical section of dorsal ectodermal cells of 48 hr gastrula stage embryo. (**D–G**) Images demonstrating anti-Sp-Efn immunoreactivity is principally associated with basal cytonemes. (**D**) Spectrin localizes to apical junctional complexes (arrows). (**E**) Sp-Efn localizes to thin filamentous projections (**c**) underlying the dorsal ectoderm. (**F**) DIC image in which the basal surface (arrowheads) of the ectodermal cells is bright relative to the rest of the cytoplasm, showing that the cytonemes project into the blastocoel. Note that most of the Sp-Efn immunoreactivity is in the cytonemal layer, not in the cytoplasm of the ectodermal cells. Bar = 15 μm (**H–J**) Sp-Efn localizes to actin containing basal projections of ectoderm. (**H**) F-actin (Alexa 488-phalloidin) is abundant in the sub-apical

*Figure 2 continued on next page*

*Figure 2 continued*

cytoplasm of ectoderm (open arrowhead) and junctional complexes (arrowheads). In addition basal projections (arrows) are also fluorescent. (I) Sp-Efn localizes to the same basal projections that bind phalloidin (arrows). Bar = 15 µm (K, L) To determine if there are membrane-bounded cytonemes on the basal surfaces of blastocoelar cells, eggs injected with RNA encoding membrane GFP (mGFP) were live imaged at 4 frames per min for intervals of 45–60 min. Individual thin, fluorescent processes appear to extend from the ectodermal cells (arrows). Bar = 4 µm See supplemental data, *Video 6*. (M, N) To determine if ectodermal cells extend actin containing cytonemal processes, eggs injected with RNA encoding Lifeact-GFP (*Riedl et al., 2008*) and live imaged at 4 frames per min for intervals of 45–60 min. In addition to processes emanating from blastocoelar cells, there are thin processes, about 15 µm in length that extend from ectoderm into the blastocoel. Bar = 4 µm See supplemental data, *Video 7*.

---

inhibitors and details of receptor activation indicate that clustering is an essential part of the activation process (*Davis et al., 1994*; *Himanen et al., 2010*; *Seiradake et al., 2010*). Our data show that Sp-Efn is abundant on cytonemal processes on the basal surfaces of dorsal ectodermal cells. This is a distinctive mechanism of Ephrin presentation and may indicate that the epithelial cells could expand their range of Ephrin signaling by distances of up to 15 µm from the cell surface. An implication of this manner of ligand presentation is that cells within the length of cytonemes could potentially activate receptors on a pigmented immunocyte without making direct contact with the cell – a form of paracrine signaling. Live imaging indicates that immunocytes migrate from the vegetal plate and contact the basal surface of the ectoderm as they migrate. Presentation of Ephrin on cytonemes suggests an active presentation of ligand in a context where immunocyte precursors move through a shag carpet of Ephrin. However, at this time there is no experimental evidence supporting a need for Sp-Efn expression on cytonemes in order for it to function. Cytonemes on the ectoderm of sea urchin embryos may have additional roles in presentation of ligands and receptors.

Thin filopodia have not previously been termed cytonemes in sea urchin embryos. *Andrews (1897)* described filopodia on the basolateral membranes of the epithelial cells of blastulae of sea urchin embryos that extend, retract, branch and fuse. He suggested that the processes are connections between blastomeres and provided cytoplasmic continuity. The existence of these

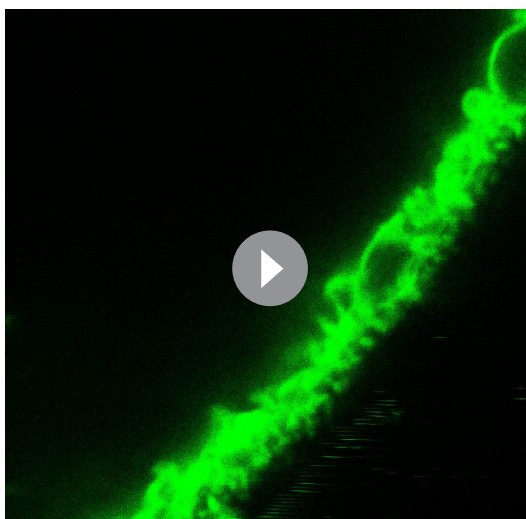

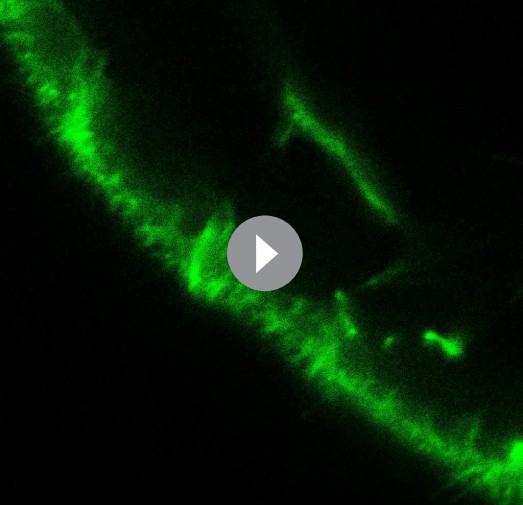

**Video 6.** Membrane GFP live imaging of dorsal ectodermal cytonemes. Live imaging of membrane GFP expressing ectodermal cells. There are numerous thin, membrane-bounded processes extending and retracting from the basal surface of dorsal ectodermal cells. (30 min sequence).

**Video 7.** Lifeact-GFP live imaging of dorsal ectodermal cytonemes. Live imaging of ectodermal cells in an embryo expressing Lifeact: GFP. There are numerous thin, actin containing processes extending and retracting from the basal surface of dorsal ectodermal cells. (30 min sequence).

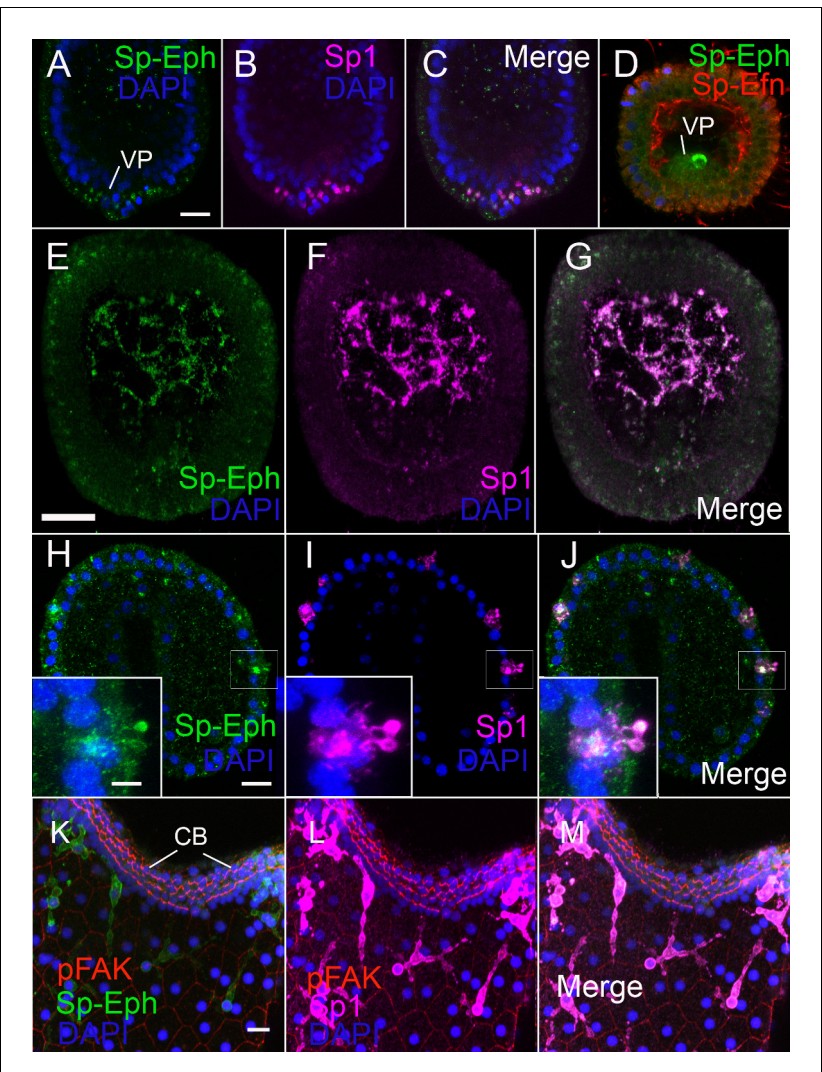

**Figure 3.** Pigmented immunocytes express Sp-Eph as they differentiate in the vegetal plate and throughout their migration and insertion into ectoderm. (**A–C**) In blastulae, SpEph can be detected in cells beginning to express the pigmented immunocyte marker Sp1. The cells are in the vegetal plate and immunoreactivity is strongest in foci adjacent to or overlapping with foci of Sp1 immunoreactivity (MeOH fixation). (**D**) At stages in which cells expressing Sp-Eph are releasing from the vegetal plate, Sp-Efn can be detected on process on the basal surfaces of ectodermal cells. Here a single pigmented immunocyte progenitor is emerging from the vegetal plate (PEM fixation). (**E–G**) Pigmented immunocyte precursors expressing surface Sp1 in the blastocoel also express Sp-Eph (MeOH fixation). (**H–J**) In gastrulae, pigmented immunocytes expressing Sp-Eph have inserted into the ectoderm. Inset images indicate that processes of immunocytes that extend through the ectoderm express Sp-Eph (MeOH fixation). See supplemental data *Video 5.* (**K–M**). In early larvae immunocytes within the ectoderm continue to express Sp-Eph on their surface (MeOH fixation). Bars = 15 μm

fine processes was confirmed with phase contrast imaging, time-lapse, and transmission EM (*Dan, 1960*; *Gustafson, 1961*; *Vacquier, 1968*). The filopodia of blastomeres were suggested to function in cell adhesion or to maintain cytoplasmic continuity of blastomeres. *Miller et al. (1995)* described ectodermal filopodia using DIC optics and noted that they are shorter and less abundant than filopodia of mesenchyme. *McClay (1999)* distinguished thin filopodia from other forms of filopodia and emphasized the potential role they may have as structures detecting patterning information. In *Drosophila*, thin filopodia have been demonstrated to be involved in signaling; either in presentation of ligands or receptors, and are distinguished by calling them cytonemes (*Ramirez-*

 

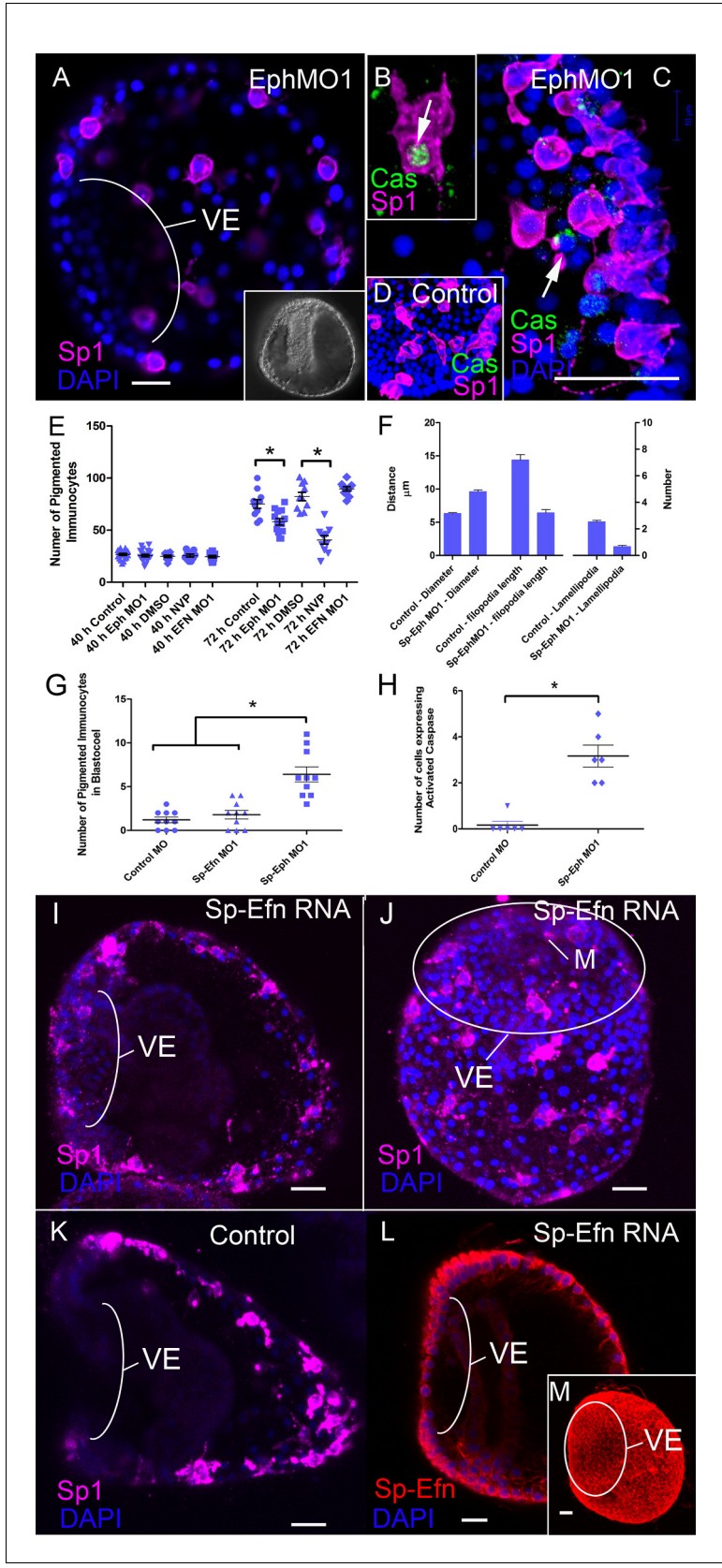

**Figure 4.** Interfering with expression of Sp-Eph or inhibition of Eph kinase function impedes immunocyte insertion into the ectoderm and they become immunoreactive to anti-Caspase3. (**A**) Maximum intensity projection of an

*Figure 4 continued on next page*

*Figure 4 continued*

embryo injected with Sp-Eph MO1 (MeOH fixation). Pigmented immunocytes are dispersed, with some having inserted into the ectoderm. The immunocytes are more rounded and do not have their typical dendritic form. They are commonly within the blastocoel. The ventral ectoderm (VE) remains clear of immunocytes. Inset: DIC image of another specimen showing the overall healthy appearance of Sp-Eph morpholino injected embryos (MeOH fixation). (**B,C**) When Sp-EphMO1-injected embryos are prepared with an antibody that recognizes only the cleaved, or activated form of Caspase3, pigmented immunocyte precursors in the blastocoel are immunoreactive (MeOH fixation). (**D**) Control morpholino injected embryos have almost no pigmented immunocytes that are anti-Caspase3 immunoreactive (MeOH fixation). (**E**) Counts of the number of Sp1 immunoreactive cells indicate that there are no differences in the number of cells among treatments (NegControlMO, Sp-EphMO1, Sp-EfnMO1, DMSO, or NVP) in prism stages. However, in early plutei there are fewer pigmented immunocytes in Sp-EphMO-injected embryos, or embryos treated with NVP. (**F**) Interfering with expression of Sp-Eph blocks the transition to epithelial-inserted, dendritic morphology. In Sp-EphMO1 injected embryos, immunocytes have larger diameters, shorter filopodia, and fewer lamellipodia than NegControlMO injected embryos. (**G**) There are fewer pigmented immunocytes in the blastocoels of 72 hr embryos injected with control MO, or Sp-Efn MO1 than there are in 72 hr embryos injected with Sp-Eph MO1. (**H**) Preparations of Morpholino injected embryos with an antibody that recognizes the activated form of Caspase3 show that there are significantly more pigmented immunocytes in the blastocoel expressing activated Caspase3. (**I–L**) Expressing Sp-Efn throughout the embryo results in mislocalization of immunocytes. (**I**) Lateral view of an embryo injected with 200 ng/µl Sp-Efn RNA. The image is a projection of 8–1 µm optical sections centered on the mouth. Note that Sp1 labelled immunocytes have inserted, or are closely associated with the ventral ectoderm (VE) (MeOH fixation). (**J**) Maximum intensity projection of the ventral surface of an embryo injected with RNA encoding full length Sp-Efn. Pigmented immunocytes are inserted into the ventral ectoderm (MeOH fixation). See *Figure 4—figure supplement 1* for a set of orthogonal projections at various levels through the image stack used to prepare this projection, which demonstrates that the Sp-1 labelled immunocytes are inserted in, or closely associated with the ventral ectoderm. The position of the oval outlining the ventral ectoderm was determined by the higher nuclear density of the cliary band. M; mouth K. Lateral view of an uninjected, control embryo prepared in the same manner as I (MeOH fixation). The image is a projection of 8–1 µm optical sections centered on the mouth. Note that Sp1 labelled immunocytes have not inserted in ventral ectoderm (VE). (**L**) A single mid-sagittal, optical section of an embryo injected with Sp-Efn RNA, showing immunoreactivity in the ventral ectoderm. Note that at this stage there is almost no expression of Sp-Efn in endoderm and the expression in ventral ectoderm appears graded (PEM fixation). (**M**) Maximum intensity projection of an embryo injected with Sp-Efn RNA showing ectopic expression of Sp-Efn in the ventral ectoderm (PEM fixation). * indicates significantly different outcomes Bars = 10 µm.

The following source data and figure supplement are available for figure 4:

**Source data 1.** Source data for *Figure 4E–H*.

**Figure supplement 1.** Orthogonal views of *Figure 4J*, a maximum intensity projection of an embryo expressing Sp-Efn throughout the ectoderm.

---

*Weber and Kornberg, 1999*). Numerous examples of signaling are now known to involve the subset of filopodia called cytonemes (*Gradilla and Guerrero, 2013*; *Kornberg, 2014*; *Kornberg and Roy, 2014*). The demonstration that thin ectodermal filopodia of sea urchin embryos present the ligand Sp-Efn indicates that they are cytonemes that function in presentation of patterning information.

The expression of Sp-Eph on pigmented immunocyte precursors, the expression of Sp-Efn on basal cytonemes, and the pattern of abundance of Sp-Efn provide a strong correlative basis for the hypothesis that Eph/Ephrin signaling is a component of the mechanisms determining the distribution of pigmented immunocytes. The principal effect of suppressing expression of Sp-Eph or inhibiting Eph kinase function is the pigmented immunocytes not adopting a characteristic dendritic morphology. The treatments result in a small increase the number of pigmented immunocyte precursors in the blastocoel, which suggests attachment to the ectoderm is independent of Eph/Ephrin signaling. Suppressing expression of Sp-Eph or inhibiting Eph kinase function results in a 20% reduction in the number of pigmented immunocytes in 72 hr embryos. There is no difference in the number of pigmented immunocytes at 48 hr when Eph is either knocked down or inhibited and activated caspase-

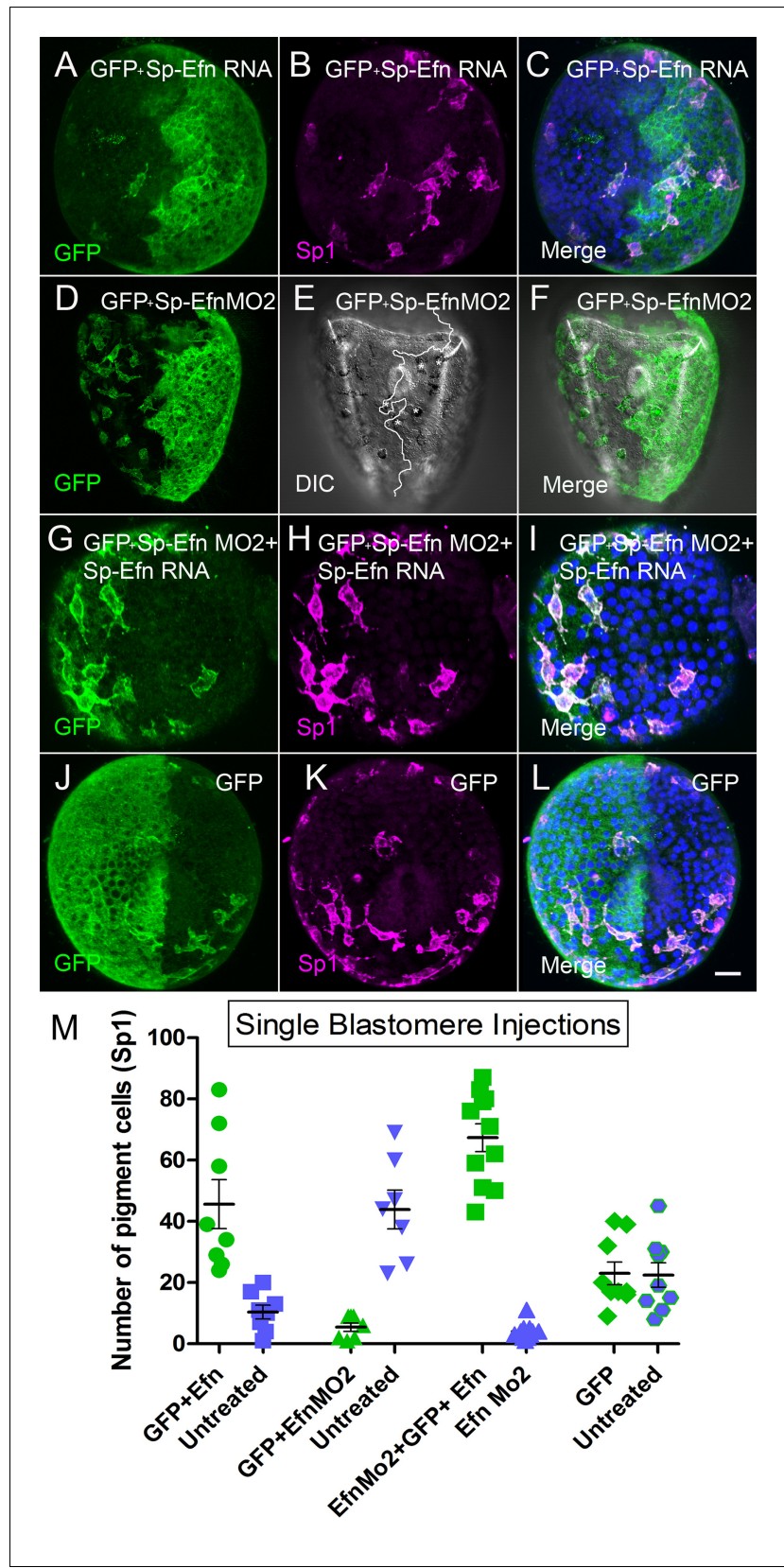

**Figure 5.** Altering the abundance of Sp-Efn in half embryos by injecting a single blastomere of 2-cell embryos indicates pigmented immunocytes insert preferentially in ectoderm expressing Sp-Efn and high levels of

*Figure 5 continued on next page*

*Figure 5 continued*

expression of Sp-Efn enhances pigment cell insertion. (**A–C**) Maximum intensity projection of an embryo in which half of the specimen co-expresses membrane GFP and Sp-Efn (MeOH fixation). Sp1 reveals the distribution of immunocytes. Note that a subset of the immunocytes expresses GFP. Most of the immunocytes are either inserted in the half expressing Sp-Efn or extend contacts to that half of the embryo. (**D–F**) Through focus projection of a living embryo that was co-injected in one blastomere with GFP and SpEfn MO2. In the DIC image immunocytes can be identified by their pigment and the white line demarks the interface between ectoderm containing morpholino (anatomical left) and untreated ectoderm (anatomical right). A subset of the immunocytes express GFP. Several pigmented immunocytes are associated with the interface between the to domains of ectoderm, those marked with * project processes to the untreated, Sp-Efn expressing, ectoderm. (**G–I**) Embryo from an egg that was injected with Sp-EfnMO2 to suppress Sp-Efn expression throughout the embryo (MeOH fixation). Once the egg had cleaved, one blastomere was injected with Sp-Efn RNA. The pigmented immunocytes are almost exclusively inserted in the half of the embryo expressing Sp-Efn. (**J–L**) Control embryos were injected with mGFP RNA only and the distribution of pigmented immunocytes can be seen to be unaffected (MeOH fixation). (**M**) Quantification of the distribution of pigmented immunocytes from the experiments depicted above (**A–L**). For each treatment (X axis) there is an injected half (green) and an uninjected half (blue). The number of pigmented immunocytes inserted in ectoderm of the injected half, or the uninjected half was determined for each embryo (72 hr). Bar = 10 µm

The following source data is available for figure 5:

**Source data 1.** Source data for *Figure 5M*.

---

3 and morphological indications of apoptosis occur in a small proportion of the immunocytes. As well, apoptosis appears to occur relatively late in the process of immunocyte migration (72 hr) and is almost completely absent in control embryos. We have concluded that the 20% reduction in the number of pigmented immunocytes is in part due to immunocytes becoming apoptotic. If Eph functioned solely as a survival factor selecting among randomly dispersed immunocyte precursors, apoptosis would be detected in control embryos. As well, using Sp1 as a differentiation marker, it is apparent that immunocyte precursors are distributed throughout the dorsal ectoderm before the end of gastrulation, which indicates patterning arises during migration. Although there is a single ligand for Eph in urchins, morpholinos directed at the receptor result in a reduction of pigmented immunocytes, and morpholinos directed at the ligand (SpEphrin) have no effect on cell numbers. This outcome may result from other interactions, both extra- and intra-cellular, with Sp-Eph, or from differences in efficiencies of the morpholinos that target the receptor or the ligand. Complete suppression of either the ligand or the receptor is necessary to fully explain this observation. Our loss-of-function data support an essential role for Eph/Ephrin signaling in the insertion of mesenchyme into epithelium, however we are unable to distinguish the precise step in the transition requiring Eph activation. The morphology of cells changes as they insert into the ectoderm, but the underlying mechanisms are not clear. Loss-of-function data does not distinguish between a sequential process and a process that

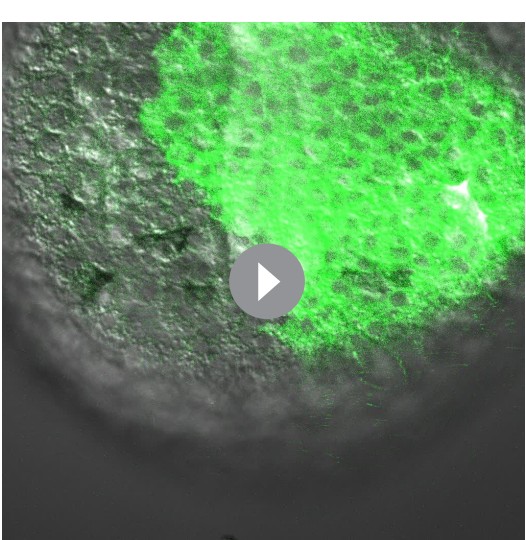

**Video 8.** Behavior of pigmented immunocyte in the context of mosaic Sp-Efn expression. In this 90 min sequence half of the embryo is expressing high levels of Sp-Efn, as indicated by the regions co-expressing GFP. Pigmented immunocytes have inserted in the ectoderm adjacent to the interface. Once inserted they do not move toward the region expressing higher levels of Sp-Efn. This sequence supports a model in which pigmented immunocytes migrate and then insert in the ectoderm without making extensive subsequent movements within the ectoderm.

is directly dependent on Eph/Ephrin. In one situation, insertion and the change in morphology are independent pathways that are sequentially activated, and only one pathway is dependent on Eph/Ephrin signaling. An alternative is that insertion and the change in morphology are independent processes, but they are directly dependent on Eph/Ephrin signaling.

By injecting Sp-Efn RNA we are able to express ligand throughout the embryo, including the ventral ectoderm, where this ligand is not normally expressed. The ventral ectoderm does have cytonemes, at similar densities to dorsal ectoderm, but they do not express Sp-Efn. Expression of Sp-Efn in ventral ectoderm is sufficient for pigmented immunocytes to disperse beneath ventral ectoderm and insert in the epithelium. As well, a high level of Sp-Efn expression in half of an embryo enhances immunocyte insertion in dorsal ectoderm. Overall, loss-of-function and gain-of-function outcomes indicate that Sp-Eph and Sp-Efn are components of the pathways that facilitate insertion of immunocytes into epithelium.

It is also clear from our data that there are additional factors that influence the distribution of Sp-Efn and the ability of pigmented immunocytes to insert in an epithelium. When Sp-Efn is ectopically expressed, early gastrula stages have uniform expression of Sp-Efn. However, by the completion of gastrulation, Sp-Efn cannot be detected in endoderm (*Figure 4L*). This may account for pigmented immunocytes not inserting into endoderm or coelomic epithelia, however the mechanism by which Sp-Efn is excluded from the endoderm is not known.

The accumulation of pigmented immunocytes in regions of abundant Sp-Efn and experiments in which Sp-Efn levels are perturbed in half of the embryo indicate that pigmented immunocytes move into regions expressing high levels of Sp-Efn. Several mechanisms could potentially account for the apparent attraction of immunocyte precursors to ectoderm expressing Sp-Efn. Given the established role for Eph/Ephrin signaling in regulating cellular adhesion, we propose a mechanism in which Sp-Eph activation leads to a localized increase in adhesion. Although soluble attractants cannot be eliminated as possible co-factors, haptotaxis may be sufficient to account for the observations that we report. Haptotaxis guides by means of a gradient of adhesion and is dependent upon adhesive bonds that are continuously made and broken. Stronger adhesions resist disruption better than weaker adhesions, with the result that random motions will move a cell up an adhesive gradient (*Davies, 2005*). Our data indicate that Ephrin abundance is not uniform and pigmented immunocytes continuously extend and retract short processes. If Eph activation leads to a localized enhancement of adhesion, random movements of cells would cause them to disperse and move to regions in which the level of Ephrin is higher. Eph activation has been demonstrated to promote integrin mediated cell adhesion in mammalian cells (*Huynh-Do et al., 1999*) and the kinase NIK (Nck-Interacting Ste20 Kinase) is activated by Eph receptors, which leads to integrin activation (*Becker et al., 2000*). As well, Eph receptors have been demonstrated to interact with Ig Superfamily receptors, which are also expressed by pigmented immunocytes (*Barsi et al., 2015*). Gradients of Ephrin have been previously demonstrated to be critical to retinotectal and corticospinal patterning. However, the predominant mechanism is for high levels of ligand to suppress projection of axons (*Suetterlin et al., 2012*; *Triplett and Feldheim, 2012*). Thus, our data suggest that Sp-Eph and Sp-Efn may also function in the dispersion of immunocyte precursors from their site of origin, in addition to their role in insertion into the ectoderm.

Phylogenetic analyses suggest that Eph receptors and ephrin ligands diverged into A- and B-types at different points in their evolutionary history, such that primitive chordates likely possessed an ancestral ephrin-A and an ancestral ephrin-B, but only a single Eph receptor (*Drescher, 2002*; *Mellott and Burke, 2008*). No extant groups of deuterostomes are known to have retained this primitive condition. However, urchins have the simplest set: a single receptor and ligand. In urchins, Eph and Ephrin are expressed in broad domains of ectoderm, where they function at the interface to mediate apical contractility (*Krupke and Burke, 2014*). We now add to this a role in mediating aspects of dispersal and epithelial transition of migratory cells. Summaries of the diverse functions of Eph and Ephrin in vertebrates emphasize broad categories of functions in establishing domains within epithelia such as we find in developing hindbrains, somites, intestinal villi, and vascular remodelling (*Klein et al., 2009*, *2012*; *Pasquale, 2005*, *2008*; *Poliakov et al., 2004*). In addition, Eph and Ephrin have a second broad set of functions in guiding migrating cells, or axonal outgrowths (*Nievergall et al., 2012*; *Klein, 2004*; *Triplett and Feldheim, 2012*: *Suetterlin et al., 2012*). This parallel suggests that urchin embryos are a model that reveals a basal range of Eph and Ephrin

 

functions, which has expanded and diversified such that Eph and Ephrin are the principal molecular determinants of cellular positioning and tissue patterning in vertebrates.

## Materials and methods

### Embryo culture and injection

Eggs and sperm were collected from *S. purpuratus* adults that had been induced to spawn by shaking. Sperm was diluted 1:100 in filtered seawater prior to fertilization and embryos were grown at 12–14°C. Eggs were prepared for microinjection as described previously (*Krupke et al., 2014*). Injection solutions contained water, 125 mM KCl and either RNA or morpholinos. Ephrin was targeted by injecting 2–4 pL of injection solution containing morpholino antisense oligonucleotides (GeneTools) against SpEphrin (Sp-EfnMO1, 2) at 400 µM. Ectopic expression of Sp-Efn was achieved using synthetic, capped mRNAs derived from the full length *S. purpuratus* Sp-Efn gene cloned in pCS2+ and transcribed using the SP6 mMessage mMachine kit (Ambion). 2–4 pL of a 0.2 µM injection solution was injected into each freshly fertilized egg as previously described (*Krupke and Burke, 2014*). Single blastomere injections followed *Krupke et al. (2014)*. Eph inhibitor, NVP BHG 712 (NVP) was used at 1.75 µM (Cat. No. 4405, Tocris Biosciences) (*Martiny-Baron et al., 2010*).

### Plasmids and reagents

Oligonucleotide DNA primers were obtained from Operon. Sequences encoding full-length Ephrin were obtained from Echinobase (http://www.echinobase.org/Echinobase/, RRID:SCR_013732) and DNA was amplified with high fidelity PCR from cDNA isolated from 72 hr *S. purpuratus* embryos and cloned using the pGEM-T Easy system (Promega, RRID:SCR_006724). Morpholino antisense oligonucleotides were obtained from GeneTools. To control for morpholino specificity, we used a control, irrelevant morpholino, which would have no effect on phenotype, as well as two experimental morpholinos against different target sequences that produced the same aberrant phenotype for each gene. Antibodies were used to confirm loss of immunoreactivity and in the case of the Sp-Efn MO rescue experiments are reported in which eggs were injected with an Sp-Efn morpholino and one cell of a 2-cell embryo injected with Sp-Efn RNA. The morpholinos targeting different sites in Sp-Efn and Sp-Eph were used interchangeably in experiments.

Morpholino sequences:
Sp-EfnMO1: 5'-AAATTTAGTCCTGGAAAGATGAGAC-3'.
Sp-EfnMO2: 5'CTCCAGGGTCAAAGTGCTCAGGTAT-3'.
Sp-EphMO1: 5'ATTGGAAAGAGTAAATCCGAGATGT-3'.
Sp-EphMO2: 5'AAATAAGTCATTCTCTCCTCTCCGT-3'.
NegControlMO: 5'GAATGAAACTGTCCTTATCCATCA-3'.

### Immunofluorescence microscopy

*S. purpuratus* embryos were collected at the desired time point and fixed for 20 min in modified PEM buffer, paraformaldehyde sea water (PFA) (*Krupke et al., 2014*) or 5 min in ice-cold methanol. Embryos were washed with PBS, blocked for 1 hr in SuperBlock (Thermo), probed with primary antibody, and washed three times with PBS. Alexa Fluor fluorescent secondary antibodies (Invitrogen, Carlsbad, CA) were used to visualize antibody labeling on a Zeiss 700 LSM (Carl Zeiss) confocal microscope. Imaging and analysis was conducted using ZEN software (Carl Zeiss). ImageJ, ZEN Lite or Adobe Photoshop (RRID:SCR_002078) was used to adjust image contrast and brightness and for final editing.

Antibodies employed have all been described previously, or are commercially developed. Data on antigens and specificity are available in the cited references.

Sp1, hybridoma supernatant diluted 1:2 (*Gibson and Burke, 1985*; Developmental Studies Hybridoma Bank, RRID:SCR_013527)

Sp-Efn, hybridoma supernatant diluted 1:2 (*Krupke and Burke, 2014*)

Sp-Eph, Rat polyclonal antiserum diluted1:400 (*Krupke and Burke, 2014*)

pFAK, (Phospho-FAK pTyr397 Antibody #44-624G, RRID:AB_2533701) Invitrogen, diluted 1:2000 (*Krupke and Burke, 2014*)

Hnf6, Rat polyclonal antiserum, diluted 1:600 (*Yaguchi et al., 2010*)

Cas3 (Cleaved Caspase-3 (Asp175) (5A1E) Rabbit mAb #9664L. Cell Signaling Technologies, diluted 1:250 (*Wei et al., 2012*).

Spectrin diluted 1:500 (*Fishkind et al. 1990*)

Alexa 488 Phalloidin (Actin Green, GeneCopia Inc.) diluted 1:250

GFP Goat pAb #ab6673, AbCam, diluted 1:500

## Image quantification

The quantification of the distribution of pigmented immunocytes and Sp-Efn immunoreactivity are described in *Figure 1—figure supplement 1*. For experiments in which one blastomere was injected, embryos were fixed and prepared with anti-GFP and Sp1 and a complete confocal, through focus series was prepared. Embryos were selected for cell counting if the GFP occupied approximately half of the ventral ectoderm, so as not to bias the surface area, which was assumed to be equal for the two halves of the embryo. Individual sections were examined sequentially and the number of immunoreactive cells (Sp1) was determined. In some embryos, through focus series were made of live embryos and the number of cells with pigment granules on each half of the embryos was determined.

## Live imaging

Live images were made using the time series acquisition features of the Zeiss LSM 700 (Zen 2009, ver. 6.0.0.303). Embryos were pipetted onto NewSilane Adhesive Coated Slides (Newcomer Supply Ltd.) and trapped under a glass coverslip attached along two edges with double-sided adhesive tape (3M Inc). Paraffin oil was applied to the open edges of the coverslip to reduce evaporation and the room temperature was controlled to 16°C. Embryos could routinely be maintained for 6 to 8 hr before they moved out of the field of view. Stacks were prepared using ImageJ (RRID:SCR_003070)

## Acknowledgements

We thank Andy Ransick, Eric Davidson, and Brad Shuster for generously providing reagents and unpublished data. Andy Ransick provided essential technical advice.

## Additional information

### Funding

| Funder | Grant reference number | Author |
|---|---|---|
| Natural Sciences and Engineering Research Council of Canada | 2413-2009 | Robert D Burke |
| University of Victoria | | Robert D Burke |
| Natural Sciences and Engineering Research Council of Canada | 2016-03737 | Robert D Burke |

The funders had no role in study design, data collection and interpretation, or the decision to submit the work for publication.

### Author contributions

OAK, Developed materials, Concepts and approaches, Performed experiments, Analyzed data, Prepared parts of the manuscript, Contributed unpublished essential data or reagents; IZ, DOM, Developed materials and performed experiments, Analysis and interpretation of data, Drafting or revising the article, Contributed unpublished essential data or reagents; RDB, Developed concepts and approaches, Performed experiments, Analyzed data, Prepared parts of the manuscript, Contributed unpublished essential data or reagents

### Author ORCIDs

Robert D Burke, http://orcid.org/0000-0001-5527-4410

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
