## [Decision Letter]

Thank you for submitting your article "Eph and Ephrin function in Dispersal and Epithelial Insertion of Pigmented Immunocytes in sea urchin embryos" for consideration by *eLife*. Your article has been reviewed by three peer reviewers, and the evaluation has been overseen by Marianne Bronner as the Reviewing Editor and K VijayRaghavan as the Senior Editor.

The reviewers have discussed the reviews with one another and the Reviewing Editor has drafted this decision to help you prepare a revised submission.

Summary:

The authors investigate the mechanisms by which embryonic pigment cells migrate to specific regions of ectoderm during sea urchin gastrulation. They demonstrate that both pigment cell precursors and differentiated pigment cells express the ephrin receptor (Eph) beginning at early gastrula. Using an antibody the authors show that the ephrin ligand (Efn) is expressed in aboral ectoderm and is absent from oral ectoderm which lacks pigment cells. Expression is on the basal side of the epithelium and is concentrated in cytoneme-like structures. Using Eph inhibitors, morpholino antisense perturbation of receptor and ligand, and mRNA misexpression the authors demonstrate that Eph signaling is necessary for proper pigment cell insertion into the ectoderm. In the absence of Eph, the pigment cell precursors remain rounded and undergo apoptosis. This is an interesting twist on Eph function as it has usually been seen a negative migratory factor that repulses contact. Overall the results presented clearly implicate Eph/ephrin signaling in the process of pigment cell patterning.

However, some of the claims are overstated and not supported by data. The most significant of these is Figure 4 which is the most important figure in the paper as it is the one that should show ectopic pigment cells in the ventral ectoderm if the hypothesis is supported, but the figure cannot be interpreted that way as presented. Next most significant is the claim that Ephrin is selectively expressed in the cytonemes or at the tips of the cytonemes.

Essential revisions:

1) Eph/ephrin terminology is a little confusing so it would be helpful to explicitly define the two in one place in the Introduction (the information is there – it would just help if it was consolidated to eliminate confusion between Eph and Efn). Also there is mention of diversification in vertebrate ephrins. A very short explanation of the numbers of genes and the designation of A and B types in relation to the sea urchin genes would add perspective.

2) Several times in the text there is mention of "mesenchymal to epithelial" transition. The pigment cells insert into the epithelium but are they really epithelial? They remain fairly motile and don't seem to completely associate with the epithelium.

3) In the beginning of the Discussion, Efn expression on the cytonemal process is said to suggest a paracrine function. Why paracrine? Some further explanation would be useful.

4) In the mRNA injection experiments, the lack of pigment cell insertion into endodermal or coelomic pouch epithelia suggests other mechanisms of control. This could be explored in the Discussion.

5) On the abscissa of the Figure 4 graph the order of treatments is slightly different between the left (40h) and right (72h) sides (Efn MO1). A congruent arrangement would reduce confusion.

6) Is there a reason why Efn MO1 was used in Figure 4 but Efn MO2 was used in Figure 5?

7) In Methods 2.1 pL injection seems like a pretty specific volume. Is this correct?

8) The authors are not correct in stating that this is the first time ectodermal cytonemes (filopodia) have been described in sea urchins. Miller and McClay (1995) identified these structures, analyzed their motility by time-lapse imaging, and discussed their possible role in signaling. The authors need to reinterpret their analysis of cytonemes in light of these previously published findings.

9) The authors emphasize that there is a single ephrin ligand and one receptor in sea urchins. Since that's true, why does the ephrin MO not produce the same effects as the receptor MO (or NVP treatment)? The Efn MO had no effect on apoptosis (PC number) at 72 hr (Figure 4) or on the number of cells that insert into the epithelium (Figure 4) but the Eph MO has significant effects in both cases. The data shown in Figure 5 argue that the Efn MO is effective, so why are there these differences between the ligand and receptor knockdowns?

10) The authors document asymmetric distributions of PCs under a variety of experimental conditions. The implication is that directional cell migration causes the asymmetry, but one could imagine scenarios where differential cell survival and/or proliferation contribute. Is it possible that the effects of Eph knockdown on PC migration are a secondary consequence of the activation of an apoptotic program, rather than the other way around? Is it possible that the asymmetric distribution of PCs seen in the experiments shown in Figure 5 is due to enhanced survival or proliferation of those PCs that happen to find themselves on the side of elevated Eph signaling? The authors should discuss explicitly why they don't think differential survival/proliferation contribute to the asymmetric distribution of PCs, especially since they document a role for Eph in apoptosis/survival of PCs.

11) Video 8 is supposed to show that pigment cells don't move much after insertion, but this looks like a pluteus stage embryo and one would have to look much earlier. This is a very important point because the PC asymmetry could come from asymmetric ingression (most pigment cells ingress from the dorsal side of the vegetal plate), directional migration within the blastocoel prior to insertion, and/or directional migration within the plane of the ectoderm.

12) One widely used control for MO specificity is to show that two different (largely non-overlapping) MOs directed at the same mRNA produce the same effect. Here the authors list two different MOs for each target (Efn and Eph) in the Methods, but for both mRNAs all the data in the paper come from a single MO the second MO is never discussed.

13) Some concerns about the image quantification shown in Figure 1: First, the authors need to include a diagram illustrating how the regions were mapped out for quantification, as this was hard to glean from the text. How did the authors insure that each region contained the same number of cells (versus black background), which would affect the average pixel intensity over the entire area? Also, the authors point out that the curves in G and H have similar shapes, but one wonders how dependent the shape of the pixel intensity curve (H) is on the specific image processing conditions that were chosen, i.e., contrast settings.

14) The authors should discuss the available evidence concerning the specificity of NVP.

15) The specificity of the antibodies is never addressed. If they have been used in previous publications, the authors should briefly summarize the data indicating that these reagents are specific, since so much of this paper relies on immunostaining.

16) Do VE cells lack filopodia entirely or do they have filopodia that are free of Efn?

[Editors' note: further revisions were requested prior to acceptance, as described below.]

Thank you for submitting your article "Eph and Ephrin function in Dispersal and Epithelial Insertion of Pigmented Immunocytes in sea urchin embryos" for consideration by *eLife*. Your article has been reviewed by two peer reviewers, and the evaluation has been overseen by a Reviewing Editor and K VijayRaghavan as the Senior Editor. The following individuals involved in review of your submission have agreed to reveal their identity: Jonathan Rast (Reviewer #1); David R McClay (Reviewer #2).

The reviewers have discussed the reviews with one another and the Reviewing Editor has drafted this decision to help you prepare a revised submission.

Summary:

This revised version of a paper from Krupke and Burke shows that pigment cells enter ectoderm that contains ephrin and do not enter ectoderm that is devoid of ephrin. They show that the pigment cells express Eph and with morpholinos show that both components are necessary for this behavior. As such this is an important advance in understanding how morphogenesis works in this embryo and serves as a model for understanding how it might work in other organisms as well.

Major Point:

1) What is meant by the sentence in the third paragraph of the Discussion "Complete suppression[…]" Is it possible that the receptor responds to inputs other than the Eph ligand? Is incomplete KO of the Eph ligand the cause of the different phenotype?

---

## [Author Response]

*Summary:*

The authors investigate the mechanisms by which embryonic pigment cells migrate to specific regions of ectoderm during sea urchin gastrulation. They demonstrate that both pigment cell precursors and differentiated pigment cells express the ephrin receptor (Eph) beginning at early gastrula. Using an antibody the authors show that the ephrin ligand (Efn) is expressed in aboral ectoderm and is absent from oral ectoderm which lacks pigment cells. Expression is on the basal side of the epithelium and is concentrated in cytoneme-like structures. Using Eph inhibitors, morpholino antisense perturbation of receptor and ligand, and mRNA misexpression the authors demonstrate that Eph signaling is necessary for proper pigment cell insertion into the ectoderm. In the absence of Eph, the pigment cell precursors remain rounded and undergo apoptosis. This is an interesting twist on Eph function as it has usually been seen a negative migratory factor that repulses contact. Overall the results presented clearly implicate Eph/ephrin signaling in the process of pigment cell patterning.

However, some of the claims are overstated and not supported by data. The most significant of these is Figure 4 which is the most important figure in the paper as it is the one that should show ectopic pigment cells in the ventral ectoderm if the hypothesis is supported, but the figure cannot be interpreted that way as presented. Next most significant is the claim that Ephrin is selectively expressed in the cytonemes or at the tips of the cytonemes.

The most important changes we have made relate to claims noted by the editorial team as lacking support. The claim that ectopic expression of Sp-Efn results in pigmented immunocytes inserting into ventral ectoderm is a critical finding and we have provided additional data to support this claim. Figure 4 now contains mid-sagittal optical sections of experimental and control embryos to demonstrate the presence and location of ectopic immunocytes (Figure 4). We have retained the original image, a ventral projection, but have added Figure 4—figure supplement 1. This supplement contains 3 orthagonal projections from the same stack of images as the ventral projection. The reviewers were concerned that the immunocytes that appear to be ventral in the projection may be located deeper in the embryo and may not be inserted in the ventral ectoderm as we claim. The orthogonal projections enable the reader to examine each pigment cell to determine that it is indeed inserted or contacting the ventral ectoderm. Thus, we have added evidence to support the claim of ectopic pigment cells resulting from ectopic expression of Sp-Efn.

The second most significant claim that one of the reviewers questioned was the expression of Sp-Efn on cytonemes. The reviewers were concerned that without a basal membrane marker, it was possible that the Efn was within the cell, not on cytonemal projections. We have added 2 panels to Figure 2 to support our claim that expression of Sp-Efn is on cytonemes. We are unaware of any good immunological markers for basolateral domains in urchin ectoderm, but F-actin and Spectrin are excellent apical markers and we have used these in conjunction with DIC to show the main cell body of the ectodermal cells, providing the reader with an image showing the basal membrane of ectodermal cells and the Ephrin-containing cytonemes extending from these cells. In the new Figure 2 the reader should be able to see that the filaments to which Sp-Efn localizes lie within the blastocoel and are not within the thickness of the epithelium. The new Figure 2 shows there is sub-apical actin and actin associated with epithelial junctions. In addition, these images show F-actin and Sp-Efn co-localize within basal cytonemes. Thus, we have provided additional evidence to support our claim that Sp-Efn is expressed on basal cytonemes of dorsal ectoderm in sea urchin embryos.

Essential revisions:

1) Eph/ephrin terminology is a little confusing so it would be helpful to explicitly define the two in one place in the Introduction (the information is there – it would just help if it was consolidated to eliminate confusion between Eph and Efn). Also there is mention of diversification in vertebrate ephrins. A very short explanation of the numbers of genes and the designation of A and B types in relation to the sea urchin genes would add perspective.

The reviewer requested that we define and consolidate receptor and ligand terminology and enumerate and describe vertebrate diversification and terminology. A brief section describing the Eph and Ephrin nomenclature and the diversity of vertebrate genes has been added to the Introduction (Introduction, para 2) and we have ensured the urchin orthologues are named and the naming is used consistently.

2) Several times in the text there is mention of "mesenchymal to epithelial" transition. The pigment cells insert into the epithelium but are they really epithelial? They remain fairly motile and don't seem to completely associate with the epithelium.

The reviewer questioned use of the terminology of “mesenchyme to epithelial transition”. The reviewer is correct in noting that the transition of pigmented immunocytes into the epithelium has not been characterized and the cells have several features that are distinct from epithelial cells. We have used the terminology of “inserting” into the epithelium to describe the process, rather than mesenchyme to epithelial transition, which implies aspects of the process that are unknown.

3) In the beginning of the Discussion, Efn expression on the cytonemal process is said to suggest a paracrine function. Why paracrine? Some further explanation would be useful.

The reviewer requested a fuller explanation of paracrine effects of cytonemal expression of Sp-Efn. We have expanded the description of the model of paracrine effects of cytonemal expression of Sp-Efn to clarify what is meant by a paracrine effect (Discussion, para 1).

4) In the mRNA injection experiments, the lack of pigment cell insertion into endodermal or coelomic pouch epithelia suggests other mechanisms of control. This could be explored in the Discussion.

The reviewer asked that we explore the lack of pigment cell insertion into endoderm and coelomic epithelia. We have added a paragraph to the Discussion (Discussion, para 5).

5) On the abscissa of the Figure 4 graph the order of treatments is slightly different between the left (40h) and right (72h) sides (Efn MO1). A congruent arrangement would reduce confusion.

The reviewer asked that we reorder the treatments in Figure 4 and we have done so.

6) Is there a reason why Efn MO1 was used in Figure 4 but Efn MO2 was used in Figure 5?

The reviewer asked if there is a reason that the one morpholino was used in some experiments and the second morpholino was used in others. A note was added to the methods stating that morpholinos producing identical phenotypes were used interchangeably. Note that in point 12, the reviewer suggests the contrary idea that we did not use both morpholinos as often as we could.

*7) In Methods 2.1 pL injection seems like a pretty specific volume. Is this correct?*

The reviewer questioned the precision of the injection volume. We have corrected to statement to show the range of volume used in injections.

*8) The authors are not correct in stating that this is the first time ectodermal cytonemes (filopodia) have been described in sea urchins. Miller and McClay (1995) identified these structures, analyzed their motility by time-lapse imaging, and discussed their possible role in signaling. The authors need to reinterpret their analysis of cytonemes in light of these previously published findings.*

The reviewer asserts that we are not correct in stating cytonemes have not previously been described and cites Miller et al. 1995. We have added a more complete review of ectodermal filopodia (Discussion, para 2) noting that filopodia have a long history of study in urchins. We distinguish cytonemes from other forms of filopodia; cytonemes function in signaling and bear either receptors or ligands. A novel aspect of our paper is that we know of no prior reports ascribing a ligand presentation function to urchin filopodia.

9) The authors emphasize that there is a single ephrin ligand and one receptor in sea urchins. Since that's true, why does the ephrin MO not produce the same effects as the receptor MO (or NVP treatment)? The Efn MO had no effect on apoptosis (PC number) at 72 hr (Figure 4) or on the number of cells that insert into the epithelium (Figure 4) but the Eph MO has significant effects in both cases. The data shown in Figure 5 argue that the Efn MO is effective, so why are there these differences between the ligand and receptor knockdowns?

The reviewer asks why there is a difference in the outcomes of receptor and ligand knockdown. We have added a brief section in the discussion (Discussion, para 3) stating that this is not resolvable without a complete ablation of either the ligand or the receptor.

10) The authors document asymmetric distributions of PCs under a variety of experimental conditions. The implication is that directional cell migration causes the asymmetry, but one could imagine scenarios where differential cell survival and/or proliferation contribute. Is it possible that the effects of Eph knockdown on PC migration are a secondary consequence of the activation of an apoptotic program, rather than the other way around? Is it possible that the asymmetric distribution of PCs seen in the experiments shown in Figure 5 is due to enhanced survival or proliferation of those PCs that happen to find themselves on the side of elevated Eph signaling? The authors should discuss explicitly why they don't think differential survival/proliferation contribute to the asymmetric distribution of PCs, especially since they document a role for Eph in apoptosis/survival of PCs.

The reviewer is concerned that the data showing ectopic pigmented immunocytes in embryos injected with Sp-Efn RNA is insufficient. As noted in comments to the editor, we have supplemented this figure with a lateral optical section and provided a supplementary figure (Figure 1—figure supplement 1) using orthogonal views to show that pigmented immunocytes are in, or closely associated with the ventral ectoderm. We have also noted in the figure that the circle indicating the location of the ventral ectoderm is defined by the higher density of nuclei in the ciliary band.

11) Video 8 is supposed to show that pigment cells don't move much after insertion, but this looks like a pluteus stage embryo and one would have to look much earlier. This is a very important point because the PC asymmetry could come from asymmetric ingression (most pigment cells ingress from the dorsal side of the vegetal plate), directional migration within the blastocoel prior to insertion, and/or directional migration within the plane of the ectoderm.

The reviewer is concerned that we do not have a single video showing the complete ontogeny of pigmented immunocytes. There are technical reasons that make this a difficult request. Keeping larvae alive and healthy on a microscope slide for 48 h is not possible at this time. We have made a series of videos to show each of the phases of immunocyte ontogeny and presented those as an alternative. The point of Video 8 is that in early plutei, pigmented immunocytes do not move extensively; they have inserted in ectoderm and stopped migrating. In *S. purpuratus*, this occurs between 60 and 72 h, which is the stage we have used to make this video. The reviewer goes on to suggest several mechanisms that may underlie the patterning of immunocytes, which is correct.

12) One widely used control for MO specificity is to show that two different (largely non-overlapping) MOs directed at the same mRNA produce the same effect. Here the authors list two different MOs for each target (Efn and Eph) in the Methods, but for both mRNAs all the data in the paper come from a single MO the second MO is never discussed.

The reviewer is concerned that we do not use the non-overlapping morpholinos to the same extent in our studies. See 6 above.

*13) Some concerns about the image quantification shown in Figure 1: First, the authors need to include a diagram illustrating how the regions were mapped out for quantification, as this was hard to glean from the text. How did the authors insure that each region contained the same number of cells (versus black background), which would affect the average pixel intensity over the entire area? Also, the authors point out that the curves in G and H have similar shapes, but one wonders how dependent the shape of the pixel intensity curve (H) is on the specific image processing conditions that were chosen, i.e., contrast settings.*

The reviewers have requested a fuller explanation of image quantification. We have added Figure 1—figure supplement 1, which contains a detailed explanation and a diagram of how quantification was done.

*14) The authors should discuss the available evidence concerning the specificity of NVP.*

The reviewer requested a discussion of NVP specificity. We have referred readers to the discussion in the paper describing NVP (Martiny-Baron et al. 2010).

*15) The specificity of the antibodies is never addressed. If they have been used in previous publications, the authors should briefly summarize the data indicating that these reagents are specific, since so much of this paper relies on immunostaining.*

The reviewer requests a summary of data on antibody specificity. As there are no novel antibodies reported, the details of antigens employed and specificity are available in the cited papers. We have added a note to the Methods explaining this (Materials and methods, para 4).

16) Do VE cells lack filopodia entirely or do they have filopodia that are free of Efn?

Do VE cells lack filopodia or do they have filopodia that are free of Efn? We have added a statement to the Discussion (Discussion, para 4) answering this question.

[Editors' note: further revisions were requested prior to acceptance, as described below.]

*Major Point:*

*1) What is meant by the sentence in the third paragraph of the Discussion "Complete suppression…." Is it possible that the receptor responds to inputs other than the Eph ligand? Is incomplete KO of the Eph ligand the cause of the different phenotype?*

The reviewer wants a clearer statement regarding the difference between the effects of the morpholino targeting the receptor and the morpholino targeting the ligand. We have altered the text of the manuscript to make this clearer. This sentence has been added: “This outcome may result from other interactions, both extra- and intra-cellular, with Sp-Eph, or from differences in efficiencies of the morpholinos that target the receptor or the ligand.”